# Women in Top Management: Performance of Firms and Open Innovation

**Safdar Husain Tahir [1,*], Muhammad Rizwan Ullah [1], Gulzar Ahmad [2], Nausheen Syed [3] and Alia Qadir [4]**

[1]  Lyallpur Business School, Government College University Faisalabad, Faisalabad-38000, Pakistan;
     mrizwanullah77@gmail.com
[2]  Department of Psychology, Lahore Garrison University, Lahore-54792, Pakistan; gulzar818@gmail.com
[3]  Department of Business Administration, Government College Women University Faisalabad,
     Faisalabad-38000, Pakistan; nausheen@gcwuf.edu.pk
[4]  Department of Management Sciences, Riphah International University Faisalabad Campus,
     Faisalabad-38000, Pakistan; alia.sheeraz@gmail.com
*   Correspondence: drsafdargcuf@gmail.com

**Abstract:** The lack of women's presence in firms' top management positions reflects gender equity problems, especially in South Asia, including Pakistan, and contours a firm's financial behavior. Based on the underpinning of the conceptual framework developed by a combination of fourteen femininity theories, the current study investigates women's induction in top management and its impact on a firm's financial behavior. We collected data from annual reports of 60 non-financial firms listed at the Pakistan Stock Exchange (PSX) for 2013–2019. The study uses the return of assets (ROA), firm's stability (FSTB), and risk-taking behavior (RTB) as dependent variables. Meanwhile, board gender diversity (BGD), female CEO (FCEO), female director-general (FDG), and female in audit committee (FIAC) are taken as independent variables. A multiple regression diagnostics approach is applied to analyze the data. The study reveals the positive impact of BGD on ROA and FSTB. However, this effect is adverse to RTB. The FIAC shows a positive (negative) impact on ROA (RTB). It also finds a negative impact of FCEO and FDG on ROA and FSTB.

**Keywords:** women; top management; risk; stability; performance; feminism; feminist theories

## 1. Introduction

Femininity has received substantial attention from researchers during the previous decades. A more significant number of empirical researches investigate business cases' gender diversity by focusing on women directors' relationship and firm performance [1,2]. Although these studies provide mixed results, some studies find beneficial impacts while some studies evidence no effect or even negative effect. Ref. [3] claims that women directors do not necessarily enhance firm profitability, but their presence may increase the board's monitoring function. They posit that women on boards serve on monitoring the board committees, and their existence enhances board meetings' overall attendance. Outcomes of women on boards provide a new dimension to the debate on femininity. These studies are now trying to provide new quantitative pieces of evidence that the firm profitability like earnings quality [4], reporting grade [5], stock price informativeness [6], and agency cost [7] improves. This research aims to categorize women directors as one of the governance mechanisms facilitating the efficient capital market. However, the available debate overlooks firms' financial behavior by analyzing women directors' impact on economic parameters. Therefore, this study attempts to examine the effect of femininity on FFB.

FFB is the firm's capability to understand the overall impact of financial decisions on an individual's (such as country, community, family, or person) conditions and make the

right decision about managing cash, opportunities, and precautions for budget planning [8]. We can define money management as human behavior [9]. Commonly, FFB includes saving behavior, cash management behavior, and, specifically, it further contains firm performance, firm stability (FSTB), and firm's risk-taking behavior (RTB) [10]. Agency theory argues that women on top-level management bring new perceptions of the intricate issues, improving the strategy formulation process. Ref. [6] posits that female representation on boards enhances the profitability and stock price informativeness. Similarly, Ref. [2] documents that more women on boards enhance the firm value. The cases of other countries also report that women on boards have a constructive effect on financial performance (FP) in many countries, i.e., China [1], Australia [11], and Spain [12]. Given the significance of femininity for the firms and investors in the financial markets, it is vital to analyze the impact of femininity on FFB. To the author's knowledge, various researches investigate the other mechanisms of corporate governance (e.g., ownership and board structure and board independence) as determinants of firm performance [1,13] and risk management [14]. However, no one has explored the impact of femininity on FFB (in terms of FP, STB, and RTB). The current study fills this gap by examining the effect of femininity on FFB.

The resources dependency theory suggests that both genders' mixture makes firms better estimate demanding resources due to different opinions and skills [15]. Both genders' leadership styles have a large difference in their behavior; men mostly keep a "Transactional" leadership style, while females have a "Transformational" leadership style. The effective leadership style in any organization increases their output when they have better results from their collateral, which is more efficient and effective [16]. Women have more job satisfaction skills for employees than men [17]. They conclude that better leadership enhances the FP of the organization. The current study provides the way to femininity in an organization and states that there is nothing important than gender; this concept varies from country to country. For Pakistan, it is more critical due to the large difference and inequality in the sphere of life. The study breaks the hurdles for females to equally participate in organizations like men.

Women have played an essential role in Pakistan's economic development. They played a vital role in the development and prosperity of the country. Throughout history, Pakistani women have held senior administrative positions, including the Prime Minister, Federal Ministers, Leader of the Opposition, Speaker of the National Assembly, High Court and Supreme Court Judges, and Major General in the Armed Forces. Despite these facts, Pakistani women face many problems when they want to use their talents, especially in the corporate sector, because of the male-dominated society. However, today's women perform brilliantly across the country in all occupations, including non-financial companies. They play an active role in the corporate sector. This study can break down these barriers.

Similarly, Pakistan's corporate governance code also requires all the listed companies to induct at least one woman on corporate boards [18]. Inclusively, the role of females (in top management) in improving the board's monitoring quality, FSTB, RTB, FP, and present requirements of having more women on boards gives an exciting setting to uncover the effect of femininity on FFB. Accordingly, the present study aims to explore the impact of femininity on FFB in Pakistan. The study motivates the current regulatory authorities to include more women in the boards in different nations like the United Kingdom, Sweden, Pakistan, Norway, Netherland, France, and Australia [19]. For instance, AICD (Australian Institute of Company Directors) makes it necessary to have at least 30% women on corporate boards, and ASX (Australian Securities Exchange) has set this target to be achieved by the end of 2018 [20]. This study may not only be "the right but the smart" thing to do.

The study extends the available debate by contributing to boardroom gender-diversity literature in four ways. First, this study systematically explores the impact of feminin-

ity on FFB for a leading emerging nation. A more significant number of empirical researches investigate business cases' gender diversity by focusing on the relationship between women directors and FP [1,2]. This study explores FFB by adding two variables: firm stability (FSTB) and risk-taking behavior (RTB). To the author's knowledge, various researches [1] have investigated the other corporate governance mechanisms (e.g., ownership and board structure and board independence). Still, few studies have explored the impact of femininity on FFB in Pakistan. Second, the study adds to the current debate by focusing on the non-financial sector by addressing the main issue: do females in top management improve FSTB, FP, and RTB? Third, the analysis links the literature on femininity and FFB with the theories. Past studies on this issue present abstruse conclusions; some studies find beneficial impacts [1,21], while some of these studies evidence no effect [3] or even negative impact [2]. From this point, the current study expects to provide evidence about this linkage. Last but most important, the study comprehensively examines (by linking the findings with theories) the determining mechanism of FFB's consequences of females by adopting a multi-approach perspective.

The rest part of the study is classified as follows: the second section summarizes the previous literature (theoretical and empirical) and constructs the hypothesis, the third section provides the data and methodology used in the study, the fourth section presents the analysis and empirical results, and the last part of the study concludes the research with theoretical as well as practical implications.

## 2. Literature Review

The scarcity of females in top leadership positions in firms generates the issue of gender equity. It designates as a phenomenon for the female leaders "who managed to climb to the top as head of business firms", leading to firms' financial behavior. Ref. [22] indicates that women are risk-averse and conservative in decision-making and consider the investment as a long-term instrument. Their risk-averse behavior does not allow them to make the right investment decisions. Most of the time, they cannot understand the nature of investment devices due to a lack of knowledge and information. They follow their ancestor's attitudes in making investment decisions. Such kind of research activities are considered the "seeds of leadership" which tribute to future leadership development concerning firms' financial behavior. Ref. [23] identifies some critical features that women entrepreneurs felt they looked for to operate their organizations. They made a comparative analysis of South African and Polish female entrepreneurs. They found potential in Polish and South African female entrepreneurs considering administration an organization. Ref. [24] highlights the significance of women entrepreneurship development as the main economic factor that forms the modern economy's innovation potential, leading to stable economic growth. They found female entrepreneurs as key possible solution providers to improve current economic and social situations. Ref. [25] shows how gender equity shapes the risk-taking behavior of firms, which affects financial behavior. They conclude on average that women are risk-averse, having less appetite for risk, and take a calculated risk during financial decisions. The primitive societies around the globe force them to fight for status and gain power. However, there are more female than male caregivers. The study indicates men are more sensation-seeking personalities and enjoy taking risks as compared to women. Ref. [26] finds that experiences and personal developmental relations contribute as the antecedents to leadership identity construction of female leaders. The previous research shows contradictory views regarding the significance of female leaders' roles; hence, they need to be explored further, particularly in the South Asian region. The current study contributes by filling this gap in the existing literature.

### 2.1. Theoretical Review

The relationship between femininity and FFB is built based on 14 theories for femininity and FP (Figure A1, see Appendix A), two approaches for femininity and RTB (Figure A2, see Appendix A), and one view for femininity and FSTB (Figure A3, see Appendix A). Some theories, such as the stakeholder theory, show a positive impact on FP. Like the vase theory, some other approaches indicate negative relation, yet others like the assimilation theory show no FP impact. Similarly, the resource dependency theory and feminist theory confirm a negative link between femininity and RTB. At the same time, the social identity theory indicates a positive relation between femininity and FSTB.

### 2.2. Empirical Review

Table 1 provides a summary of empirical literature for the impact of femininity on FFB. The table shows that [27–30] found a positive effect of board gender diversity on FP. In contrast, Ref. [31] reported a negative impact of gender diversity on FP. Refs. [27,32] documented a negative effect of gender diversity on firm RTB and the positive impacts of femininity on FSTB. [33] found a positive effect of female CEO on FP; while [34] observed a negative effect of female CEO on FP. Refs. [35–38] showed a negative effect of female CEO on RTB, and [35] found a positive effect of female CEO on FSTB.

Refs. [39–43] reported a positive effect of female director-general on FP. Ref. [40] found a positive while [43] observed a negative impact of female director-general on firm RTB. Moreover, Ref. [42] presented a positive influence of female director-general on FSTB. Females in the audit exerted a negative impact on FP [44–46]. Ref. [46] also found a negative effect of females in audit committee on FSTB.

The above theoretical and empirical literature provides mixed results, which allows developing the following hypothesis:

**H1:** *Female participation in top management influences (a) firm financial performance, (b) firm risk-taking behavior, and (c) firm stability.*

**Table 1.** The literature on the impact of femininity on firm financial behavior.

| Femininity | Authors | Performance | Risk-Taking | Stability |
|---|---|---|---|---|
| Board Gender Diversity | [32] | ---- | Negative | Positive |
| | [27] | Positive | Negative | ---- |
| | [31] | Negative | ---- | ---- |
| | [28] | Positive | ---- | ---- |
| | [29] | Positive | ---- | ---- |
| | [30] | Positive | ---- | ---- |
| Female CEO | [35] | ---- | Negative | Positive |
| | [34] | Negative | Negative | ---- |
| | [45] | "No difference between the risk attitudes of male and female CEOs." | | |
| | [36] | ---- | Negative | ---- |
| | [37] | ---- | Negative | ---- |
| | [38] | ---- | Negative | ---- |
| | [33] | Positive | ---- | ---- |
| Female Director-General | [39] | Positive | ---- | ---- |
| | [40] | Positive | Positive | ---- |
| | [41] | Positive | ---- | ---- |
| | [42] | Positive | ---- | Positive |
| | [43] | Positive | Negative | ---- |
| | [46] | Positive | ---- | ---- |

|  |  |  |  |  |
|---|---|---|---|---|
|  | [44] | Negative |  | Negative |
| Female in Audit | [47] |  | No impact |  |
| Committee | [46] | Negative | ---- | ---- |
|  | [48] | Negative | ---- | ---- |

## 3. Methodology

The study analyzes the impact of femininity on FFB. The sample of the study consists of 60 non-financial firms listed at the Pakistan Stock Exchange (PSX).

We extracted annual financial reports from selected firms, which cover the period 2013–2019. The study uses regression analysis to analyze the impact of femininity on FP, FSTB, and RTB. FFB is used as an explained variable, while femininity is used as an explanatory variable. The study also uses firm size, firm age, and board size as control variables. The description and measurement of variables are shown in Table 2.

**Table 2.** Variables description.

| Variable Name | Explanation/Measurement | Source |
|---|---|---|
| **Financial Behavior** | | |
| Financial Performance (FP): Return on Assets (ROA) | ROA = Net Income/Total Assets | [49] |
| Firm Stability (FSTB) | FSTB = Ln (1 + Z)<br>Where: Z = (K/TA + ROA)/SD (ROA)<br>Where: K = Capital, TA = Total Assets, ROA = Return on Assets and SD = 3 years rolling standard deviation | [33] |
| Risk-Taking Behavior (RTB) | $RTB = \sqrt{\frac{1}{T-1}\sum_{t=1}^{T}\left(Eit - \frac{1}{T}\sum_{t=1}^{T} Eit\right)2}$<br>Where: T = 3<br>$Eit = \frac{EBITDAi,t}{ASSETSi,t} - \frac{1}{Nk,t}\sum_{j-1}^{Nk,t}\cdot\frac{EBITDAj,t}{ASSETSj,t}$<br>"Where: *i* indexes firms and t indexes year. Nk,t indexes firm numbers within industry k and year t. For each firm with available earnings and total assets for at least three years in 2013 to 2019. We estimate the firm's EBITDA/ASSETS deviation from the industry average (for the corresponding year) first. Then the standard deviation of this measure for each firm is calculated". | [1] |
| **Femininity** | | |
| Board Gender Diversity (BGD) | The proportion of Females on Board | [50] |
| Female CEO (FCEO) | "1" If CEO is Female, otherwise "0". | |
| Female Director General (FDG) | "1" If Director-General is Female, otherwise "0". | |
| Females in Audit Committee (FIAC) | No. of Females Sits in Audit Committee | [2] |
| **Control Variables** | | |
| Firm Age (FAGE) | No. of years since the firm (incorporated) | [51] |
| Size of Board (SOB) | No. of Board Members | |
| Size of Firm (SOF) | Natural Logarithms of Total Assets | |

In quantitative research, we do not rely on fully randomized experiments, and control variables are an essential vital tool to rule out rival alternative explanations for the hypnotized linkages. Our study consciously uses three control variables such as firm age (FAGE), size of Board (SOB), and size of the firm (SOF). These variables control the changing market and economic conditions, macroeconomic situation, and board members' skills and experience in the following ways. First, as shown in Ref. [45], previous studies show the strong connection between SOF and economic conditions. For example, bad economic conditions decrease profitability, which ultimately reduces the SOF or vice versa. Second, FAGE controls the gained skills and experiences of the board of directors and top management. The study does not consider the global financial crisis because no crisis was evidenced between 2013 and 2019.

*3.1 Econometric Models*

For analyzing the impact of femininity on FFB, the study uses the following econometric models:

$$ROA_{it} = \beta_0 + \beta_1(BGD_{it}) + \beta_2(FCEO_{it}) + \beta_3(FDG_{it}) + \beta_4(FIAC_{it}) + \beta_5(Controls) + E_{it}\ldots, \tag{1}$$

$$FSTB_{it} = \beta_0 + \beta_1(BGD_{it}) + \beta_2(FCEO_{it}) + \beta_3(FDG_{it}) + \beta_4(FIAC_{it}) + \beta_5(Controls) + E_{it}\ldots, \tag{2}$$

$$RTB_{it} = \beta_0 + \beta_1(BGD_{it}) + \beta_2(FCEO_{it}) + \beta_3(FDG_{it}) + \beta_4(FIAC_{it}) + \beta_5(Controls) + E_{it}\ldots, \tag{3}$$

where: ROA: Return on Assets, FSTB: Firm Stability, RTB: Risk-Taking Behavior, FCEO: Female Chief Executive Officer, BGD: Boards Gender Diversity, FDG: Female Director-General, FIAC: Female in Audit Committee, Control variables include FAGE: Firm Age, SOB: Size of Board, SOF: Size of Firm, and $\beta_0$ is a constant term while $\beta_1\ldots \beta_5$ are regression coefficients, and E is the error term.

## 4. Results and Analysis

### 4.1. Summary Statistics and Multicollinearity

Table 3 represents that the mean value of ROA is 0.03 ranging from −1.21 to 0.57. The average TQ value is 0.72, with the maximum and minimum values of 0.97 and −0.14, respectively. The mean value of FSTB (RTB) is 1.66 (0.70). The mean value of BGD in sample firms is 0.22 showing that the selected firms consist of 22% women directors, and the mean ratio of FIAC is 32% showing that 32% of females sit in audit committees. In the sample firms, 8.8% of CEOs are females. On average, the selected firms have about eight members on their board (board size), out of which 22% are female directors.

**Table 3.** Descriptive statistics.

| Variables | Mean | Median | Maximum | Minimum | Std. Dev. |
|---|---|---|---|---|---|
| ROA | 0.0320 | 0.0230 | 0.5670 | −1.2100 | 0.3250 |
| FSTB | 1.6610 | 1.5270 | 7.3750 | −2.4600 | 1.3060 |
| RTB | 0.7000 | 0.5700 | 3.1600 | 0.0300 | 0.5200 |
| BGD | 0.2200 | 0.2220 | 0.7500 | 0.0000 | 0.1490 |
| FCEO | 0.0880 | 0.0000 | 1.0000 | 0.0000 | 0.2840 |
| FDG | 1.6180 | 2.0000 | 6.0000 | 0.0000 | 1.0900 |
| FIAC | 0.3120 | 1.0000 | 3.0000 | 0.0000 | 0.7930 |
| FAGE | 31.5020 | 27.0000 | 69.0000 | 9.0000 | 12.8120 |
| SOB | 8.2292 | 4.9400 | 11.5600 | 7.0000 | 0.2900 |
| SOF | 14.8800 | 14.8800 | 18.7600 | 10.7000 | 1.5800 |

The correlation analysis is used to predict the strength of the relationship among all the study variables. Table 4 presents the Pearson Correlation Matrix for all the variables to check the multicollinearity in the data. BGD shows a positive correlation with ROA and

FSTB while negatively correlated with RTB suggesting that BGD enhances firm profitability and stability while decreasing its risk-taking. The highest correlation coefficient (0.58) is between FDG and BGD showing that multicollinearity does not affect the data.

**Table 4.** Correlation Matrix.

| Variables | ROA | FSTB | RTB | BGD | FCEO | FDG | FIAC | FAGE | SOB | SOF |
|---|---|---|---|---|---|---|---|---|---|---|
| ROA | 1.00 | | | | | | | | | |
| FSTB | 0.13 | 1.00 | | | | | | | | |
| RTB | 0.32 | 0.15 | 1.00 | | | | | | | |
| BGD | −0.18 | 0.12 | 0.11 | 1.00 | | | | | | |
| FCEO | −0.17 | −0.07 | −0.24 | 0.25 | 1.00 | | | | | |
| FDG | 0.11 | 0.09 | 0.21 | 0.58 | 0.26 | 1.00 | | | | |
| FIAC | 0.23 | 0.42 | −0.30 | 0.55 | 0.24 | 0.54 | 1.00 | | | |
| FAGE | 0.13 | −0.17 | −0.37 | −0.16 | 0.06 | −0.14 | −0.15 | 1.00 | | |
| SOB | 0.08 | −0.37 | −0.21 | −0.10 | −0.09 | 0.08 | −0.22 | 0.12 | 1.00 | |
| SOF | 0.18 | −0.15 | −0.22 | −0.17 | −0.07 | −0.15 | −0.14 | 0.18 | 0.31 | 1.00 |

*4.2. Regression Analysis*

4.2.1. Impact of Femininity on Firm Performance

Table 5 (model 1) shows a positive impact of BGD on ROA ($\beta = 0.0513$, $p \leq 0.05$). These findings suggest that a one percent increase in women directors' percentage improves FP by 5.1%. The findings are in line with the previous studies [27–30]. The results are in line with previous studies claiming that better-governed firms have higher profitability [1,52,53]. The existence of females in the audit committee (FIAC) exerts a positive impact on ROA ($\beta = 0.0498$, $p \leq 0.01$). It suggests that a one percent increase in female proportion in the audit committee increases ROA by about 5%. These outcomes are inconsistent with prior studies [45–48]. This evidence is allied with females' active participation in the audit committee meetings [3]. FCEO ($\beta = -0.0311$, $p \leq 0.05$) and FDG ($\beta = -0.0867$, $p \leq 0.05$) have negative impact on ROA. One percent increase in FCEO and FDG leads to a decline in ROA by 3.1% and 8.7%, respectively. It posits that hiring a female as CEO or having a female in the audit committee is associated with lower profitability. Therefore, H1a is fully supported. Moreover, the control variables FAGE ($\beta = 0.0006$, $p \leq 0.05$), SOB ($\beta = 0.3718$, $p \leq 0.01$), and SOF ($\beta = 0.0206$, $p \leq 0.01$) have significant positive impact on ROA.

4.2.2. Impact of Femininity on Firm Risk-Taking

In Table 5 (model 2), BGD shows a negative impact on RTB ($\beta = -0.0336$, $p \leq 0.05$). The results imply a one percent increase in women directors' percentage causes to decrease firm risk-taking by 3.4%. The results are consistent with [32]. The results report that a lower risk provides economic benefits to the firm's stakeholders (such as employees and suppliers). There is a negative impact of FIAC on RTB ($\beta = -0.0947$, $p \leq 0.01$). Increasing one percent of FIAC leads to a decline RTB by 9.4%. The evidence indicates that females in audit committees reduce risk-taking, which is associated with conservatism in strategic and risk oversight [54], strict monitoring [55], and solidity and quality of risk oversight [48]. The study could not find any significant impact of FCEO and FDG on RTB. These outcomes are inconsistent with past studies [34–38]. Hence, H1b is partially accepted. The control variables SOB ($\beta = 0.4403$, $p \leq 0.01$) and SOF ($\beta = 0.2049$, $p \leq 0.01$) are positively correlated with RTB.

4.2.3. Impact of Femininity on Firm Stability

Table 5 (model 3) reports that BGD has positive impact on FSTB ($\beta = 0.0805$, $p \leq 0.05$). The positive coefficient of BGD shows that a one percent increase in the proportion of female directors improves FSTB by 8%. The results are similar to [32]. The results are also

consistent with the social identity theory that women are not eager to take a risk, which eventually improves FSTB. The impact of FCEO ($\beta = -0.0496$, $p \leq 0.05$) and FDG ($\beta = -0.0487$, $p \leq 0.10$) on FSTB is found to be negative. It reports that a one percent increase in FCEO and FDG causes to decline the FSTB by approximately 5%.

**Table 5.** Regression Analysis.

| Independent Variables | Dependent Variable | | | | | |
|---|---|---|---|---|---|---|
| | **Model 1: ROA** | | **Model 2: RTB** | | **Model 3: FSTB** | |
| | Coefficient | *p*-Value | Coefficient | *p*-Value | Coefficient | *p*-Value |
| Constant | −0.9793 | 0.0000 *** | 12.7849 | 0.0000 *** | −0.5108 | 0.0832 |
| BGD | 0.0513 | 0.0460 ** | −0.0336 | 0.0387 ** | 0.0805 | 0.0213 ** |
| FCEO | −0.0311 | 0.0306 ** | −0.4341 | 0.5437 | −0.0496 | 0.0505 ** |
| FDG | −0.0867 | 0.0220 ** | 0.5424 | 0.3231 | −0.0487 | 0.0749 * |
| FIAC | 0.0498 | 0.0199 *** | −0.0947 | 0.0000 *** | 0.0869 | 0.2537 |
| FAGE | 0.0006 | 0.0462 ** | −0.0025 | 0.5375 | −0.0147 | 0.0013 *** |
| SOB | 0.3718 | 0.0000 *** | 0.4403 | 0.0000 *** | 0.5581 | 0.1654 |
| SOF | 0.0206 | 0.0000 *** | 0.2049 | 0.0000 *** | 0.0461 | 0.2203 |
| R² | 0.7863 | | 0.6897 | | 0.6365 | |
| Adjusted R² | 0.7435 | | 0.6437 | | 0.6037 | |
| Hypothesis | H1a: Fully Accepted | | H1b: Partially Accepted | | H1c: Partially Accepted | |

Note: *** ($p < 0.01$), ** ($p < 0.05$), * ($p < 0.10$).

Moreover, FIAC insignificantly correlates with FSTB. Therefore, H1c is also partially supported. SOB and SOF have an insignificant impact on FSTB.

The endogeneity issue is the main obstacle to understanding the nature of association different variables qualitative research of corporate finance. Most of the time, the used variables are scarce and endogenous in the relationship, making causal relation complicated. Ref. [56] suggests some ways to handle the endogeneity problem in empirical corporate finance: first, understanding the negative relationship between CEO power and firm performance. Second, to change the sign of coefficients from positive to negative. Third, instrumental variables lagged dependent variables, and fixed-effect models follow the generalized method of moments (GMM). Last, Ref. [57] decomposing the incentive of executives into time-variants. Our study results show the adverse relation of FECO with the dependent variables in all three models. Thus, there is no potential endogeneity issue in the analysis.

## 5. Discussions: Women in Top Management, Performance, and Open Innovation

Open innovation enhances business progress by allowing the organizations to influence more ideas from different external sources [58]. The essential advantage of open innovation is that it appreciates the possibility that the organizations will achieve business progress due to additional sales from new products or production technologies [59]. Refs. [60,61] also reached similar conclusions. The studies indicate a positive relationship between innovation and sales growth, which is very beneficial for its profitability. Ref. [62] suggested that the focused innovation firms are more likely to persist in the gradually turbulent world. The study concluded that organizations' long-term stable growth is highly dependent on the firm's commitment to the innovational process. Thus, the innovational productivity of the organization remains the prime concern of stakeholders. However, many researchers indicated that several agency problems are aligned with the firm's innovational productivity that is likely to be severe for the following two reasons: first, some managers are "risk-averse" who are anxious about the job security, and therefore, they reduce their investments in the firm's innovational process because they perceive such investments riskier; and instead of investing their money in the innovational

projects, they invest money in routine tasks. Second, the managers might choose a secret life and averse to the innovational projects' inflated efforts. Thus, monitoring has to be strengthened to enhance the governance of innovation [63]. The literature revealed that the female directors improve the efficiency of internal administration of invention as an excellent exemplification of female directors on board is aligned with efficient monitoring [3], increased public revelation [64], perfect board individuality and engagement, and improved board discussions of complex issues [65]. Hence, it is reasonable to believe that female directors help mitigate the agency problem and promote organizational innovation through effective monitoring. Moreover, the female directors convey different attitudes, opinions, and problem-solving skills to the board [66,67]. Hence, female directors' presence on boards enhances suggested ideas, smooth creativity, and produces more strategic substitutes [68]. Innovation can be predictable to boost the decision-making process's exhaustiveness, avoid information processing and decision-making biases, and positively affect the organizational design.

## 6. Conclusions

### 6.1. Females and Financial Behavior

Since the pattern of femininity's effect on FFB has not to be clarified to a reasonable extent. Therefore, this important study needs to explain, review, and further balance the various conclusions on the female's role in determining the firm's financial behavior. The current study proposes that the potential ways for the association between females in top-level management and the firm's economic behavior are built based on 12 challenging theoretical perspectives. The study concludes that females have different impacts on a firm's financial behavior from various theories' perspectives.

### 6.2. Theoretical Assumptions

Each theory presents its theoretical assumptions on female characteristics and environmental features. Females play their role in top management, which is, in actuality, time-dependent, industry-dependent, region-dependent, country-dependent, and even culture-dependent. However, the practical facts are that each theory's assumptions are somewhat met, which naturally results in multi-paths (multi-approach) for participating in determining the firm financial behavior held by the different theoretical perspectives that instantaneously function to the other extent. The study finds all kinds of behavioral consequences of female executives in the available empirical debate. To the authors' best knowledge, this study is the first to comprehensively examine and explain the influencing mechanism of females' participation in top-level management on firm financial behavior from 14 theories by developing a conceptual model based on a multi-approach perspective.

### 6.3. Key Findings

The study finds a positive impact of BGD on ROA. These findings suggest that a one percent increase in women directors' percentage improves FP by 5.1%. Through their oversight and monitoring ability, the study's outcomes advocate that women directors affect the firms' profitability. The results are in line with previous studies claiming that better-governed firms have higher firm profitability [1,52,53]. The existence of females in the audit committee (FIAC) also exerts a positive impact on ROA. It suggests that a one percent increase in females' proportion in the audit committee increases ROA by about 5%. This evidence is allied with females' active participation in audit committee meetings [3]. Inclusively, the shreds of evidence support the call (by Pakistani Companies Act 2017) for having at least one woman on the corporate board. The results are similar to Malaysian and Australian government policies of having 30% of females on corporate boards. FCEO and FDG harm ROA. One percent increase in FCEO and FDG leads to declining ROA by

3.1% and 8.7%, respectively, indicating that hiring females as CEOs or as director-general (DG) is associated with lower profitability. Therefore, H1a is fully supported.

BGD shows a negative impact on RTB, implying that a one percent increase in women directors causes to decrease the firm's risk-taking by 3.4%. One possible implication for this evidence is that females present more stable performance, which is in line with the hypothecation that females are highly risk-averse. This attitude towards risk is consistent with prior evidence showing that females focus more on monitoring activities and implementing strict governance. The results also report that a lower risk provides economic benefits to the firm's stakeholders (such as employees and suppliers). There is a negative impact of FIAC on RTB. Increasing one percent of FIAC leads to a decline RTB by 9.4%. The evidence indicates that females in audit committees reduce risk-taking, which is associated with conservatism in strategic and risk oversight [47], strict monitoring [54], and solidity and quality of risk oversight [55]. Here, H1b is partially accepted.

The results report that BGD positively impacts FSTB, showing that a one percent increase in female directors' proportion improves FSTB by 8%. The results are consistent with the social identity theory that women are not eager to take the risk, which eventually enhances FSTB. We found a negative impact of FCEO and FDG on FSTB. It reports that a one percent increase in FCEO and FDG causes to decline FSTB by approximately 5%. Therefore, H1c is also partially supported.

### 6.4. Theoretical Implications

The study finds a positive impact of BGD and FIAC on FP. These findings align with the stakeholder theory, resource dependency theory, catfish effect theory, agency theory, human capital theory, social capital theory, and social cognition theory. The stakeholder theory [69] determines that female executives had better meet the stakeholders' expectations, including the public, the non-profit organizations, the government, the other partner companies, the community, and the employees. These expectations make the companies' image in the stakeholders' eyes positive which is advantageous for the companies to get stakeholders' support. According to the resource dependency theory [70], one possible implication for the positive impact of femininity on FP is that females in senior leadership have different mindset resources, relationship resources, and knowledge resources. In case of shortage of resources of top-level management entirely comprised of males, FP needs to improve. Another implication is that the gender-diverse board has access to a larger pool of resources, strengthening a firm's network with its outside environment and encouraging additional perspectives and resources. The catfish effect theory [71] supports the study's evidence of females' positive role in determining a firm's FP. In line with the approach, females are catfishes to the males and have less competency. Accordingly, to prove their competency, females work harder than males, which results in a good performance. Aligning with the agency theory [72], women on top-level management bring a new perception of the intricate issues that help improve the strategy formulation process. Accordingly, the findings indicate that more women on boards enhance the firm value. The results supported by the human capital theory [73] which posits that females in top management teams have a high quality of human capital than males, which causes a positive impact of femininity on FP. The positive effect of femininity on FP also supports the social capital theory [74] that states that women on corporate boards build a robust social network. This social network ultimately improves firm profitability. The findings also support social cognition theory [75], which reports that women directors' views are only seriously taken when they reach the majority level, which improves the FP.

FCEO and FDG exert a negative impact on FP. These outcomes align with the vase theory, liberal feminism theory, social feminism theory, human capital theory, social capital theory, and agency theory. According to the vase theory of feminism [76], female participation in top management considers a "useless vase". They may be demoted and do not have extensive power to be a part of the imperative decision-making process, making females' ability to phantom and stop them from contributing more to the firm. The females

are also viewed as "vases" because they lack experience, work capacity, and have less experience than males. This result leads to the fact that females create lower FP than males. In line with the liberal and social feminism theories, FCEO and FDG hurt FP because females typically face discrimination while receiving education and business skills. Still, the males are more suitable for top executive positions. Therefore, liberal feminist theory [74] reports that females negatively influence FP. According to social feminist theory [77], females are more responsible for caring for family, and males are more accountable for earning money from jobs and work. Under this classification, females spend more time and energy on homework, which reduces their energy and time for work.

Consequently, in light of the above point of view, females contribute less than males, negatively impacting FP. The findings are also consistent with human capital theory [78], which claim that females in top management teams have a low quality of human capital than males and negatively impact FP. The social capital theory [79] holds the viewpoint that women on corporate boards have a minimal social network compared to men because they have less energy and time than men to build a robust social network that negatively impacts FP. Another possible implication for the negative impact of FCEO and FDG supports the agency theory, which argues that the existence of female directors in a firm's board enhances the effectiveness of board's monitoring which is useful to minimize the conflicts of interest between owners and top management teams. However, when the intensity of monitoring derived from female directors becomes highly strict about implementing managerial decisions, female directors' importance changes into negative [72]. Moreover, the findings are inconsistent with upper echelon theory [80]. There is a negative impact of BDG and FIAC on RTB, which is in line with the resource dependency theory and feminist theory. According to resource dependency theory [81], a significant reason for this negative impact is that non-financial firms in Pakistan are primarily dependent on female staff. Furthermore, large-sized firms facing legitimacy pressure are likely to have more females on their boards. Hence, the firms that are facing environmental constraints elect females on their boards as a means to reduce risk. Therefore, feminism exerts a negative impact on firm risk-taking behavior. Another reason for the negative effect as feminist theory [82] argues is that males are usually socialized to positively take the risk and hold masculine characteristics such as independence and aggressiveness. These characteristics ensure the success of males in the workplace.

In contrast, females are caregivers and usually prefer their private sphere, such as the home. Therefore, feminine traits discourage risk-taking. Moreover, the study also reports a positive impact of BGD on FSTB, which is in line with the social identity theory [32] that explains that women are not eager to take a risk, which eventually improves firm stability.

### 6.5. Practical Implications

As a practical implication, the study evidenced that (i) BGD improves FP through their oversight and monitoring ability, (ii) females present more stable performance, which is in line with the hypothecation that females are highly risk-averse [83], and (iii) the existence of FIAC exerts a positive (negative) impact on FP (RTB). Inclusively, the shreds of evidence support the call (by Pakistani Companies Act 2017) for having at least one woman on the corporate board and are also aligned with Malaysian and Australian government policies of having 30% females on corporate boards. Therefore, the government of Pakistan's policy of having at least one female on corporate boards and Malaysian and Australian government policies of having 30% females on corporate boards should continuously utilize and enforce to benefit from having a male and female mixture composition of boards. Pakistani non-financial firms are advised to invest in a good pool of female talent and search for qualified females who bring additional expertise to the corporate boardrooms and the audit committees. Hiring females on corporate boards and having females on the audit committee will be extremely important in the industrial sectors having fewer female candidates.

*6.6. Suggestion and Recommendations*

i.　The study recommends that having at least one female on corporate boards (by Pakistani Companies Act, 2017) should be continuously utilized and enforced by the firms to get the benefits of having a mixture of males and females in the boards' composition better financial behavior;

ii.　The firms are advised to invest in a good pool of female talent and search for qualified females who bring additional expertise to the corporate boardrooms and the audit committees;

iii.　The study suggests increasing the women ratio in the audit committee and board of directors.

*6.7. Limitations and Future Research Directions*

i.　The study's findings highlight the need to develop a more comprehensive and differentiated conceptual model based on a multi-approach perspective, integrating the potential performance, stability, and risk effects of females' representation;

ii.　Future researchers should focus on providing the practitioners with systematic and clean suggestions and tools on improving the positive consequences of a firm's financial behavior and simultaneously weaken the adverse effects of females' participation in top-level management and integrating 14 theoretical perspectives expansively;

iii.　The study uses only one measure of each dependent variable, i.e., ROA for financial performance, RTB for risk-taking behavior, and FSTB for firm stability. Future researchers may challenge the findings by using alternative measures to obtain better outcomes;

iv.　The analysis covers the data of seven years only; future studies may increase the data period to get more complete results;

v.　The study also faces difficulties in finding sample firms with female executives.

**Author Contributions:** Conceptualization, M.R.U. and S.H.T.; methodology, M.R.U.; software, G.A.; validation, G.A., N.S., and A.Q.; formal analysis, S.H.T.; investigation, A.Q.; resources, N.S.; data curation, M.R.U.; writing—original draft preparation, S.H.T.; writing—review and editing, G.A.; visualization, A.Q.; supervision, N.S.; project administration, S.H.T. All authors have read and agreed to the published version of the manuscript.

**Funding:** This research received no external funding.

**Institutional Review Board Statement:** Not applicable.

**Informed Consent Statement:** Not applicable.

**Data Availability Statement:** The data presented in this study are available on request from the corresponding author.

**Conflicts of Interest:** The authors declare no conflict of interest.

**Acknowledgments:** We thank the Government College University Faisalabad, Pakistan, for using the library and internet services to complete the research article.

**Conflicts of Interest:** The authors declare not conflict of interest.

**Appendix A**

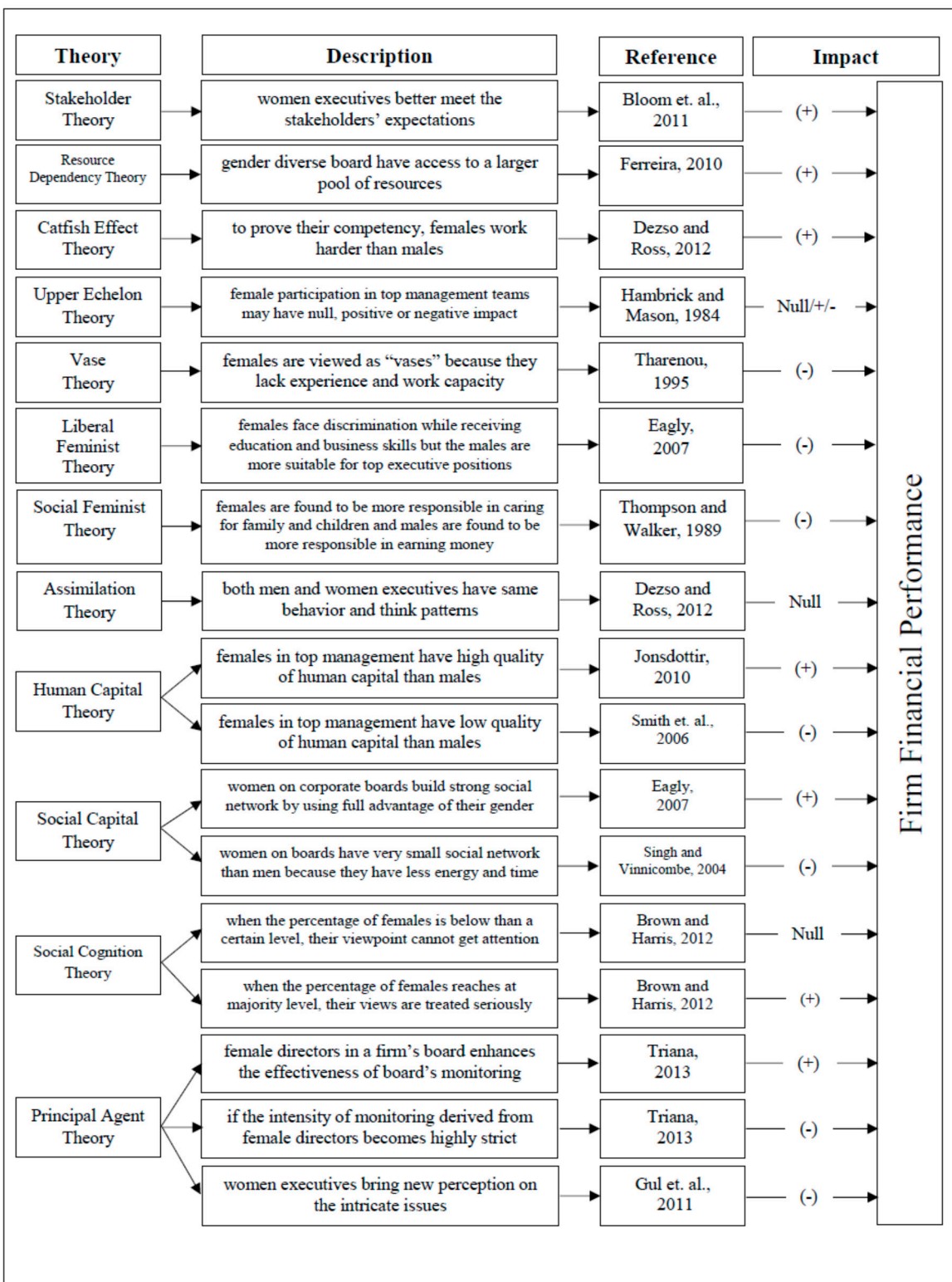

**Figure A1.** Firm performance: conceptual model based on multi-approach perspective.

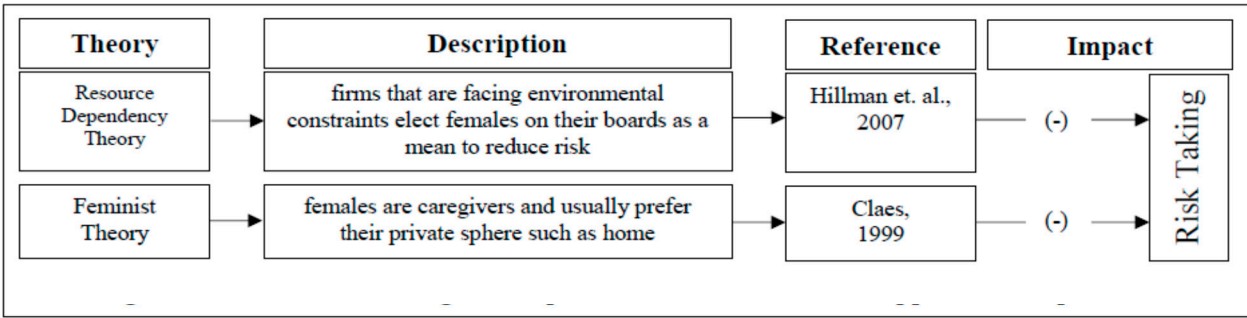

**Figure A2.** Firm risk-taking: conceptual model based on multi-approach perspective.

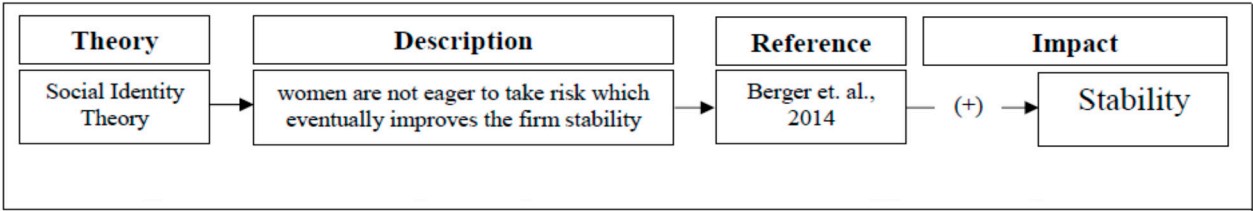

**Figure A3.** Firm stability: conceptual model based on multi-approach perspective.

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
