# Peer review of "Women in Top Management: Performance of Firms and Open Innovation"

_2199-8531, doi:10.3390/joitmc7010087_

Round 1
Reviewer 1 Report
I am very grateful to review this interesting paper. It is well written, however I suggest some major improvements.
Describe clearly the study aim and methodology in the abstract
The literature review part must be improved with some more sources, comments and hypothesis elaboration based on this as now it is very poor.
Please consider also following sources:
Imm, C.L., Wahid, N.A.mThe seeds of leadership: From the experiences of senior Malaysian women leaders. (2020) Polish Journal of Management Studies, 22 (1), pp. 200-216.
Kot, S., Meyer, N., Broniszewska, A. A cross-country comparison of the characteristics of Polish and South African women entrepreneurs. (2016) Economics and Sociology, 9 (4), pp. 207-221.
Onyusheva, I., Meyer, N. The features of female entrepreneurship development in Kazakhstan: An analytical survey. (2020) Polish Journal of Management Studies, 21 (1), pp. 265-282.
Nowak, M. Entrepreneurship in china’s greater bay area-a gender perspective (2020) Polish Journal of Management Studies, 22 (2), pp. 324-344.
Al-Tkhayneh, K., Kot, S., Shestak, V. Motivation and demotivation factors affecting productivity in public sector. (2019) Administratie si Management Public, 2019 (33), pp. 77-102
Results section must be developed with some more introduction to data presentation as well as comments and explanations.
Results must be also discussed in relation to the previous studies already published.
Author Response
Dear Reviewer 1
I hope you are well. The required changes have been made. Please see attachment.

Reviewer 2 Report
Please find attached the Review Report

Author Response
Dear Reviewer 2
I hope you are doing well. The required changes have been made in the revised paper. Please see the attachment.

Reviewer 3 Report
attached

Author Response
Dear Reviewer 3
I hope you are doing well. The required changes have been made in the revised paper. Please see the attachment.

Round 2
Reviewer 1 Report
I am satisfied of the improvements!
Author Response
Thank you very much for your comments!
Reviewer 2 Report
The authors are addressed the issues raised in the revised version.
Author Response
Thank you very much for your comments!
Reviewer 3 Report
congrats!
Author Response
Thank you very much for your comments. Improved and editing has done.